# Unpacking the V1 map: Differential covariation of preferred spatial frequency and cortical magnification across spatial dimensions

**Marc M. Himmelberg**[1,2]*, **Yuna Kwak**[1], **Marisa Carrasco**[1,2], **Jonathan Winawer**[1,2]

1 Department of Psychology, New York University, New York, New York, United States of America,
2 Center for Neural Science, New York University, New York, New York, United States of America

* marc.himmelberg@nyu.edu

## Abstract

Primary visual cortex (V1) has long served as a model system for understanding cortical organization. Although its structural and functional properties vary markedly across its surface, patterns of covariation suggest possible underlying constancies. Such constancies would imply that V1 is composed of multiple identical units whose visual properties differ only due to differences in their inputs. To test this, we used fMRI to investigate how V1 cortical magnification and preferred spatial frequency covary with eccentricity and polar angle, measured in 40 observers. V1 cortical magnification and preferred spatial frequency were strongly correlated across eccentricity and around polar angle, however their relation differed between these dimensions: they were proportional across eccentricity but not polar angle. The constant ratio of cortical magnification to preferred spatial frequency when measured as a function of eccentricity suggests a shared underlying cause of variation in the two properties, e.g., the gradient of retinal ganglion cell density across eccentricity. In contrast, the deviation from proportionality around polar angle implies that cortical variation differs from that in retina along this dimension. Thus, a constancy hypothesis is supported for one of the two spatial dimensions of V1, highlighting the importance of examining the full 2D-map to understand how V1 is organized.

## Author summary

Human primary visual cortex (V1) is the first region of the brain to receive visual input. Its organization, whether built from uniform repeating units or following more complex patterns, remains debated. We assessed this by measuring two key V1 properties: cortical magnification (how much cortex processes each region of the visual field) and preferred spatial frequency (the level of detail neurons respond best to). Using MRI, we found these properties are closely linked, but their relation differs by visual field dimension. Across eccentricity (distance

**Data availability statement:** The data generated for this study are available on OSF at https://osf.io/c8vks/. Customized vistadisp software can be found at https://github.com/WinawerLab/vistadisp.

**Funding:** This work was funded by NIH NEI grants R01-EY027401 and P30EY013079 to M.C. and J.W; the NYU Center for Brain Imaging pilot grants to M.M.H., Y.K., M.C., and J.W.; and by the NYUAD Center for Brain and Health, funded by Tamkeen under NYU Abu Dhabi Research Institute grant CG012. The funders played no role in the study design, data collection and analysis, decision to publish, or preparation of the manuscript.

**Competing interests:** The authors have declared that no competing interests exist.

from the center of gaze), the two properties decrease at a fixed ratio, reflecting structural patterns in the retina. Around polar angle (circular dimension), this proportionality no longer holds, even though the two properties still covary. These results show that whereas V1 is partly governed by a simple rule of proportionality, its organization cannot be explained as just repeating units. Instead, different spatial dimensions follow different principles, emphasizing the need to consider the full two-dimensional layout of the visual map to understand how vision is represented in the brain.

## Introduction

The properties of human cerebral cortex are linked to perception, cognition, and action. Thus, understanding how its structure and function are organized is an important goal in basic and clinical science. A useful scale for describing cortical organization is the cortical area [1]. One such area, primary visual cortex (V1), has been particularly fruitful for examining spatial organization, with many visual properties varying systematically throughout the V1 map: position tuning [2], cortical magnification [3,4], receptive field size [5], and preferred spatial frequency [6]. A more complete picture, however, should describe how V1 properties covary throughout the V1 visual field representation.

One possibility is that multiple V1 properties scale proportionally. For example, the visual field representation becomes increasingly compressed with eccentricity: cortical magnification (mm/deg) declines linearly with eccentricity, 100-fold from fovea to periphery [7]. If V1 preferred spatial frequency (cycles/degree) also declined linearly with eccentricity, the two properties would be proportional, and thus their ratio would be a constant. This would translate to a fixed number of mm of cortex per stimulus spatial frequency cycle as a function of eccentricity. This *cortical constancy* would be expected if V1 were composed of many canonical cortical circuits, each doing the same computations on the visual stimulus up to a scale factor on the image. This could be accomplished by each local circuit being the same, with neural tuning varying across locations only because of the visual information inherited from their inputs, rather than local circuits differing from one another, as in the log-polar model of V1 topography [8,9]. Identifying which properties of cortical organization are constant leads to a better understanding of how the system functions [10].

At the other extreme, there would be no relation between different V1 properties. This possibility can be ruled out, however, because multiple properties vary with eccentricity, including cortical magnification [7] and preferred spatial frequency [11–13]. However, it is unknown whether preferred spatial frequency and cortical magnification follow the same rate of decline with increasing eccentricity. Similarly, cortical magnification [14,15] and preferred spatial frequency [12] both vary with polar angle. However, it is unknown whether and how these properties covary when measured as a function of polar angle.

Further, the overall surface area of V1 –and thus overall V1 cortical magnification– varies up to 3-fold among observers [16–20]. If overall V1 cortical magnification covaries with overall preferred spatial frequency –i.e., larger V1s have higher preferred frequency– this would extend the cortical constancy to span individual differences.

Here, we first investigated how V1 preferred spatial frequency varies with eccentricity and polar angle. We captured this pattern in a 2D model, which predicts preferred spatial frequency from the full V1 map –accounting for variation as a function of eccentricity and polar angle. Next, we investigated how cortical magnification varies as a function of eccentricity and polar angle and again fit a 2D model to these data across the full V1 map. Finally, we assessed how preferred spatial frequency and cortical magnification covary throughout the V1 visual field representation and among individual observers.

Measuring covariation throughout the V1 map and among individuals requires highly reliable data in many observers. To this end, a large sample of observers (n = 40) completed two fMRI experiments, one to measure cortical magnification and one to measure spatial frequency tuning. To preview our results, we found that: First, V1 preferred spatial frequency and cortical magnification varied systematically with eccentricity and polar angle. Second, 2D models provided good fits to the preferred spatial frequency and cortical magnification data, from which we derived parametric descriptions of how these properties varied throughout V1. Third, preferred spatial frequency covaried with cortical magnification when measured as a function of eccentricity and polar angle. As a function of eccentricity, preferred spatial frequency tightly scaled with cortical magnification. As a function of polar angle, the properties covaried but were not proportional, as cortical magnification varied twice as much as preferred spatial frequency. Finally, across individual observers, the properties covaried but were not proportional, as preferred spatial frequency varied twice as much as cortical magnification. The covariation among individuals indicates that observers with greater cortical magnification (i.e., a larger V1) tend to have higher preferred spatial frequency.

## Results

### Parcellating the V1 visual field representation

First, we parcellated each observer's V1 into segments, each of which encoded a localized region of the visual field. To do so, we used fMRI to measure retinotopic maps in visual cortex of 40 observers, examples of which are illustrated in Fig 1 A. Using population receptive field (pRF) estimates [21], we parcellated each observer's V1 map into 128 segments (64 segments per V1 hemisphere; Fig 1B). To this end, we defined sixteen 22.5° angular wedges that, in total, spanned 360° around the visual field. Each of the angular wedges were composed of eight 1° eccentricity segments, centered between 2° and 9° eccentricity.

Next, we computed preferred spatial frequency and cortical magnification for each V1 segment. Preferred spatial frequency was computed using a two-stage analysis, first estimating the BOLD response amplitude to each stimulus and then fitting spatial frequency tuning curves to these data. For each observer, response amplitudes were estimated for each vertex using a GLM [22]. This yields BOLD response amplitudes for each of 20 spatial frequency stimuli (Fig 2), which were then averaged across the vertices within a V1 segment and across observers, yielding one beta weight for each stimulus per V1 segment.

Second, we fit log-Gaussian tuning curves to group-average BOLD responses, separately for the 10 pinwheel stimuli, the 10 annulus stimuli, and the averages of the two stimuli ('combined'). Group averaging was performed using a bootstrap procedure: we drew 40 samples with replacement from the 40 observers, computed the group-average, and fit log-Gaussians to the resulting data. This process was repeated 50 times.

Examples of these tuning curves are illustrated for several V1 segments in Fig 3A–3B. The log-Gaussian provided a good fit to the spatial frequency data within each segment, with a median $R^2$ of .96 across the 128 segments for the combined condition ($CI_{95}$ = [.94,.97], 50 bootstraps across observers). We then extracted the preferred spatial frequency value (i.e., the spatial frequency at which the log-Gaussian peaks) for each V1 segment from the log-Gaussian fit, averaging the data across orientation (pinwheels and annuli) for each spatial scale. This removed any orientation effects from our data.

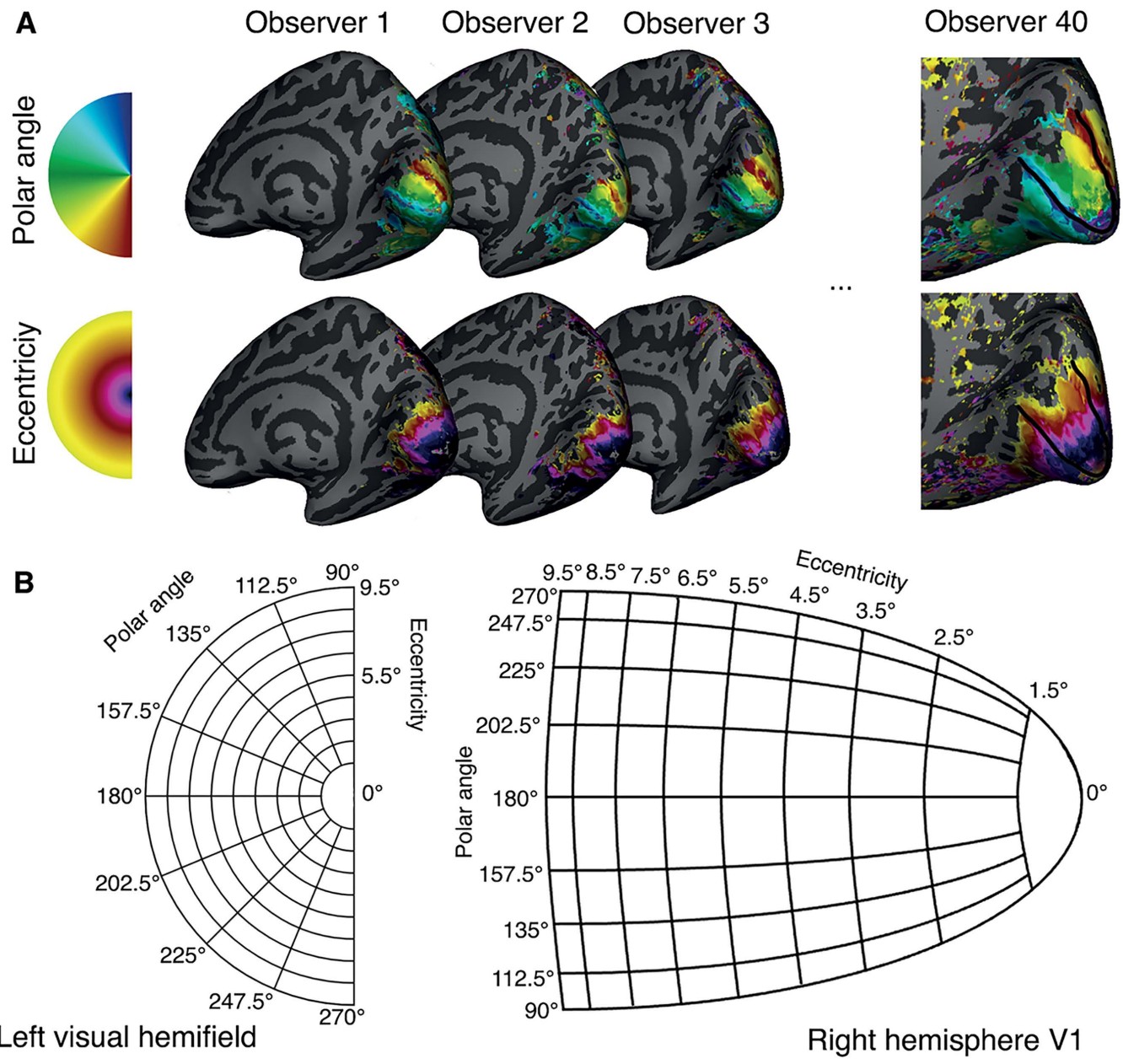

**Fig 1. Examples of retinotopic maps and schematic of V1 parcellation into visual field segments. (A)** Examples of polar angle and eccentricity maps projected onto the inflated right hemisphere for 4 observers. The 'Observer 40' mesh is zoomed in to show V1, the outline of which is illustrated by the black border. **(B)** Schematic of the parcellation of the V1 representation into eccentricity and polar angle segments for a single hemisphere. For each observer, and each hemisphere, V1 is parcellated into 64 segments (eight 22.5°-polar-angle segments and eight 1°-eccentricity segments) using the retinotopy data.

Next, for each V1 segment we quantified group-average areal cortical magnification (i.e., mm² V1 surface area/deg² visual space), again using a bootstrapping procedure. We summed the surface area of the vertices within a V1 segment (Fig 3C) and divided this value by the area of visual space (in degrees²) that the segment encoded (Fig 3A). Example measurements of cortical magnification (mm² surface area/degrees² visual space) are shown in Fig 3C.

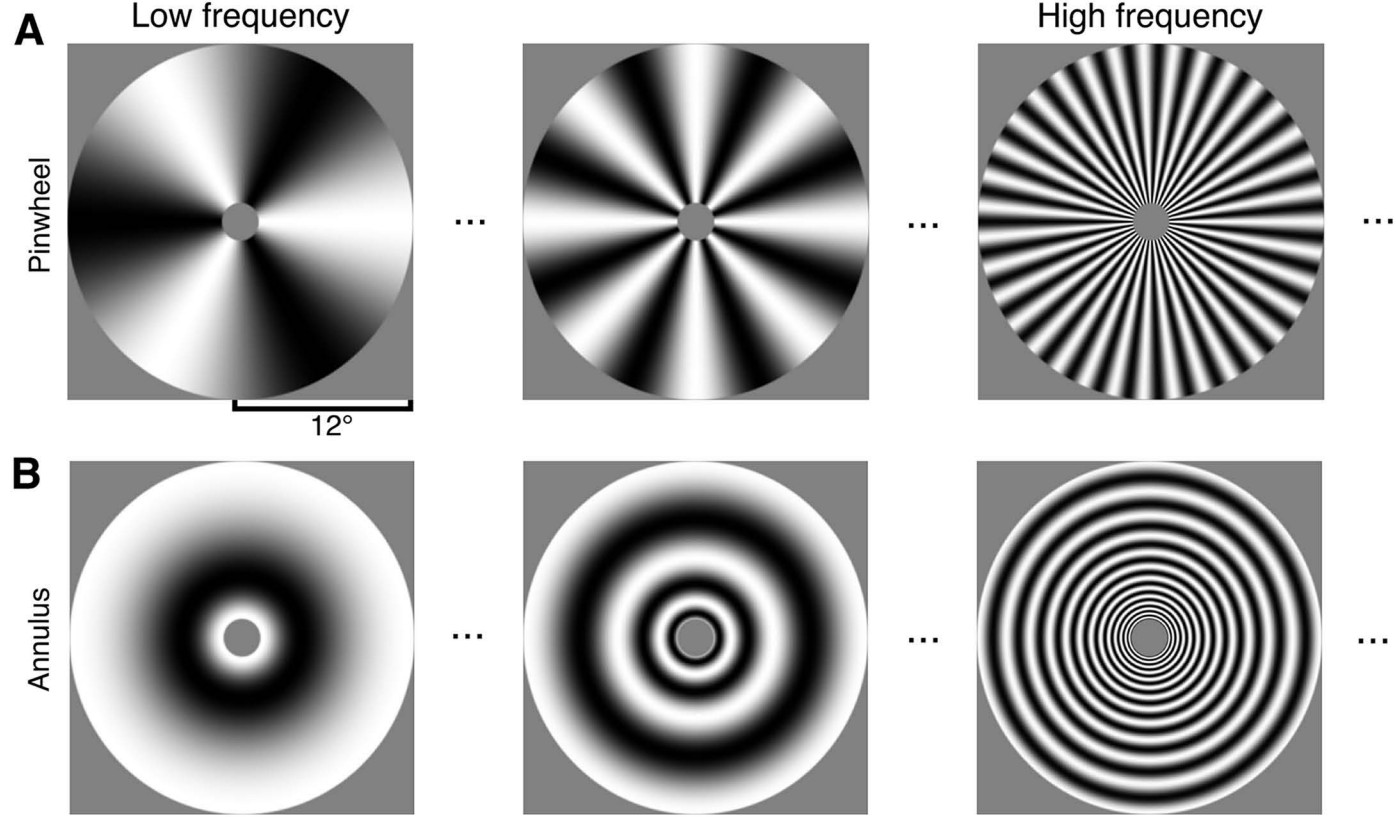

**Fig 2. Examples of (A) pinwheel and (B) annulus stimuli used for estimating V1 spatial frequency tuning curves.** These are three of the 10 spatial scales tested (base frequencies of 3, 8, and 35). The local spatial frequency of each stimulus increases with distance from the center. For the top and bottom rows, the local spatial frequency is matched at corresponding locations, but the orientations are orthogonal.

Group-averaged preferred spatial frequency and cortical magnification for each V1 segment are available in S1 and S2 Tables.

### Modeling preferred spatial frequency and cortical magnification throughout the visual field

Typical models of human V1 cortical magnification account for its variation as a function of eccentricity [7,23–30], but not polar angle. Here, we implemented a 2D model to account for properties that vary as a function of two dimensions: eccentricity *and* polar angle. We fit this model separately for preferred spatial frequency and cortical magnification, using each of the 128 V1 segments.

We made the simplifying assumption of independence for the two spatial axes: namely, that variation due to polar angle follows the same pattern at every eccentricity, and variation as a function of eccentricity follows the same pattern at every polar angle. Interactions between eccentricity and polar angle are small relative to the effects of eccentricity and polar angle (S1A and S2 Figs). Modeling the interactions would require multiple additional free parameters that could compromise reliability of the fits. Therefore, we modeled the polar angle variation as a multiplicative modulator around the average cortical magnification at each eccentricity. We use the same equation for preferred spatial frequency.

The 2D model has the general form:

$$M(r, \theta) \;=\; f(r) \;\cdot\; g(\theta) \qquad\qquad \text{Equation 1}$$

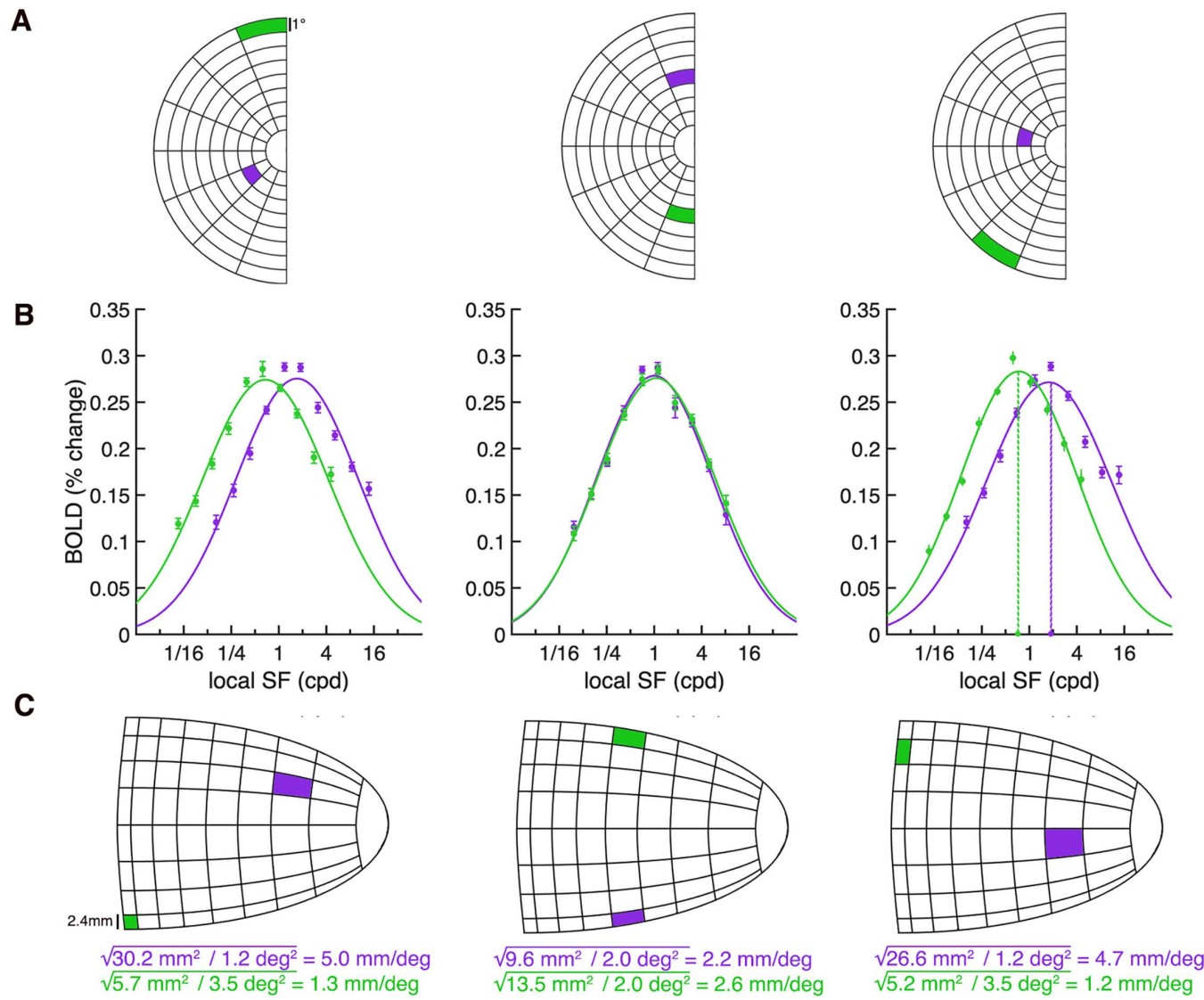

**Fig 3. Computing cortical magnification and spatial frequency tuning curves from V1 segments. (A)** Schematic of the left visual hemifield broken down into 64 V1 segments (eight 22.5° polar angle wedges, eight 1° eccentricity segments). Data in (B) and (C) are derived from vertices spatially tuned to the example locations in **(A)**. **(B)** Group-average log-Gaussian fits for example V1 segments. In the examples shown, a log-Gaussian is fit to combined stimuli data (i.e., BOLD responses averaged between pinwheel and annuli stimuli). The dashed colored lines in the third plot show the preferred spatial frequency (cpd) for two example segments. Error bars represent ±1 standard deviation (SD) across 50 bootstrapped group-averages. **(C)** To compute cortical magnification (mm/deg): the surface area of the vertices within each V1 segment (shown by the colored square in the right hemisphere V1 schematic) are summed and then divided by the amount of visual space that the segment encodes, shown by the corresponding visual field segment in **(A)**. We then take the square root of this value to compute linear cortical magnification.

Where $r$ is eccentricity, $\theta$ is polar angle, and $M$ is cortical magnification (or preferred spatial frequency). For the function of eccentricity, $f$, we adopt the inverse linear parameterization widely used for linear cortical magnification [4,7]:

$$f(r) = \frac{A}{r + B} \qquad \text{Equation 2}$$

When *A* increases, the function in Fig 4A shifts upwards, indicating greater V1 cortical magnification. When *B* decreases, the function in Fig 4A shifts to the right, indicating greater V1 cortical magnification at central V1 relative to the periphery.

For the polar angle modulator, *g*, we used the sum of two harmonics, centered at 1:

$$g(\theta) = 1 + \alpha \, cos(2\theta) - \beta \, sin(\theta)$$

Equation 3

When $\alpha$ is positive, there is greater V1 cortical magnification along the horizontal than vertical meridian; a horizontal meridian asymmetry (HVA) (Fig 4B). When $\beta$ is positive, there is greater V1 cortical magnification along the lower than upper vertical meridian; a vertical meridian asymmetry (VMA) (Fig 4 B). Larger values of $\alpha$ and $\beta$ translate to more pronounced asymmetries.

Together, the equation has four free parameters, *A*, *B*, $\alpha$, and $\beta$.

$$M(r, \theta) = \frac{A}{r + B} \cdot (1 + \alpha \, cos(2\theta) - \beta \, sin(\theta))$$

Equation 4

In sum, combining the eccentricity-dependent function with a polar angle modulator enables the model to predict how cortical magnification or preferred spatial frequency varies throughout the entire visual field (Fig 4C). The model output declines with increasing eccentricity, and at each eccentricity, varies with polar angle. Equation 4 assumes a linear rather than an areal measure, such as cortical magnification in units of mm/deg or preferred frequency in cycles/deg. Thus, we fit the models to linear measurements of preferred spatial frequency and cortical magnification.

The spatial frequency tuning curves were fit to group-averaged beta weights, separately for each of the 128 V1 segments. The group averaging was done with a bootstrap procedure, in which we drew 40 samples with replacement of the 40 observers, repeating this 50 times, giving us 50 different estimates of the group-average data. The model provided an excellent fit to the data; median $R^2 = .99$ across the 50 bootstraps ($CI_{95}$ = [.98,.99]) (Fig 5A). The model fit captured the

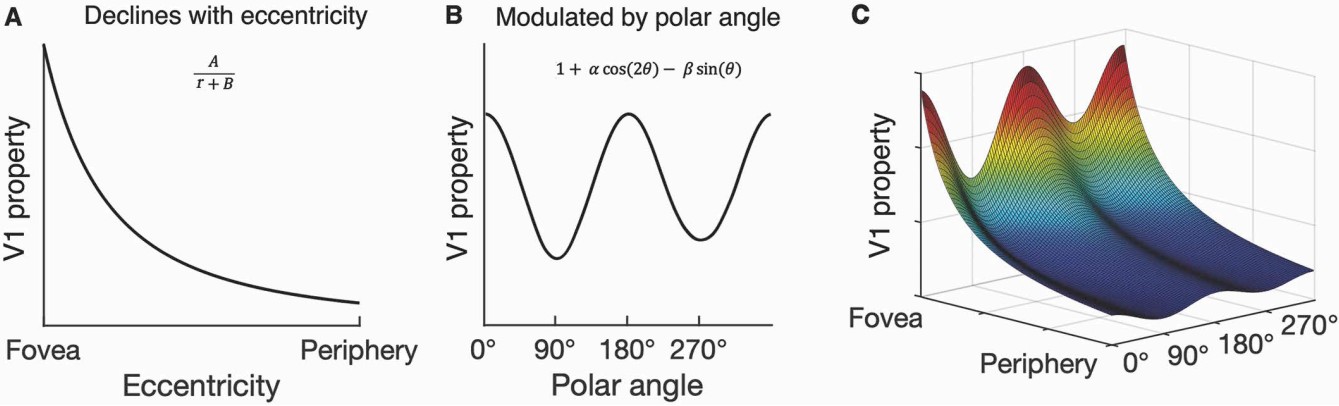

**Fig 4. Visualization of 2D model predicting how neural properties vary throughout the V1 representation. (A)** The decline in a neural property (in this case, V1 preferred spatial frequency or cortical magnification) as a function of eccentricity is described by the widely adopted parameterization from [7]. **(B)** V1 neural properties can be modulated by polar angle; the model predicts larger values along the horizontal (0°, 180°) than vertical meridian (90°, 270°), and along the lower vertical (270°) than upper vertical (90°) meridian of the visual field. **(C)** Visualization of the full model showing how a neural property varies throughout the visual field; the decline in preferred spatial frequency or cortical magnification from **(A)** is modulated by a function of polar angle from **(B)**.

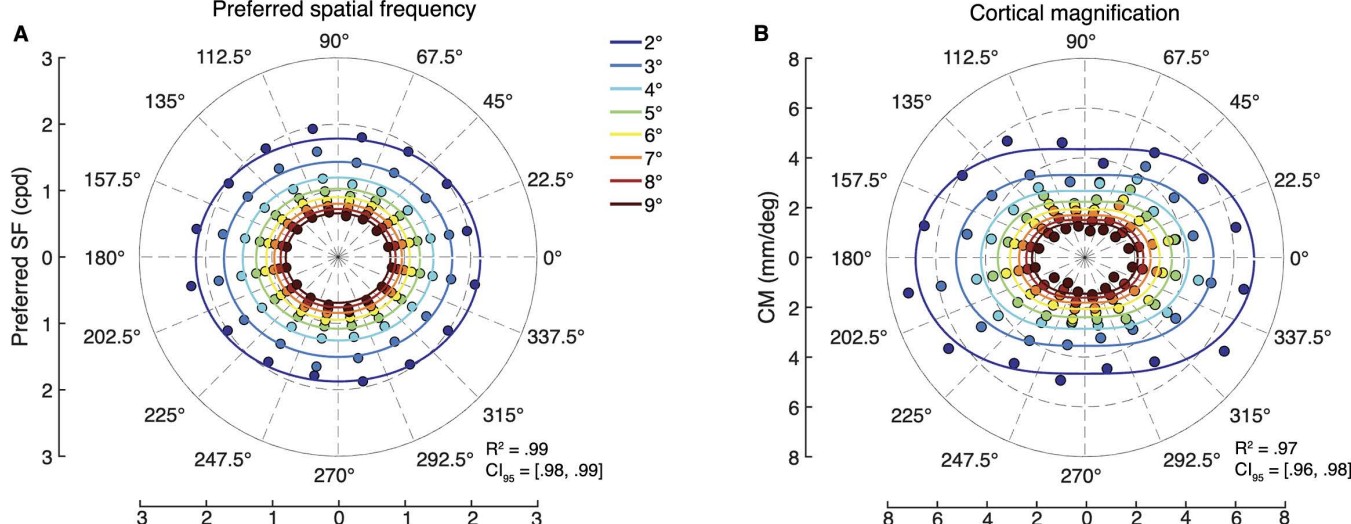

**Fig 5. Preferred spatial frequency and cortical magnification throughout the visual field.** Each data point comes from a V1 segment, and each color corresponds to an eccentricity bin. The model fit is shown as lines for each eccentricity bin, although only a single model was fit to the full range of data in each of (A) preferred spatial frequency and (B) cortical magnification.

decline in preferred spatial frequency as a function of eccentricity, and polar angle asymmetries: higher preferred spatial frequency at the horizontal than vertical meridian, and at the lower than upper vertical meridian.

The model captures the variation in preferred spatial frequency as a function location. From the log-Gaussian fits used to estimate preferred spatial frequency, we also extracted spatial frequency bandwidth, which reflects the range of frequencies a region of V1 responds to. In S3 Fig we show bandwidth as a function of eccentricity along the horizontal, lower, and upper vertical meridians, and across all of V1. Bandwidth (measured in octaves) is approximately constant along each meridian. Consequently, linear bandwidth (in cycles per deg) is approximately proportional to preferred frequency, consistent with the idea of scale-invariant neural tuning [31].

As the spatial frequency of the grating stimuli declined with eccentricity (Fig 2), we considered the possibility that estimates of the decline in preferred spatial frequency with eccentricity were due to this property of the stimuli rather than genuine differences in preferred spatial frequency. In a supplementary analysis, we compared preferred spatial frequency estimates derived from scaled gratings (as used here and in prior work [11]) with those derived from uniform gratings [12]. The three studies report similar estimates of preferred spatial frequency as a function of eccentricity, indicating that stimulus bias is unlikely to be a major factor in our parameter estimates (S4 Fig). In addition, Ha et al. [13] demonstrated through simulations that scaled gratings do not bias their model toward finding a spurious relation between preferred spatial frequency and eccentricity.

Parameter estimates from the model fit to preferred spatial frequency and from the model fit to cortical magnification data are shown in Table 1. The median preferred spatial frequency parameter estimates derived from the model fit are shown in the first row.

The 2D model also provided an excellent fit to the cortical magnification data (Fig 5B); median $R^2 = .97$ ($CI_{95} = [.96, .98]$). The model captured the decline in cortical magnification as a function of eccentricity and greater cortical magnification at the horizontal than vertical, and lower than upper vertical meridian of the visual field. The parameter estimates derived from the model fit to the cortical magnification data are shown in the second row of Table 1; the $A$ and $B$ parameters (17.9, 1.2) are in excellent agreement with Horton and Hoyt (17.3, 0.75) [7].

PLOS Computational Biology

**Table 1. Median model parameter estimates and 95% confidence intervals from the models fit to measurements of preferred spatial frequency and cortical magnification.**

**Model parameter estimates**

|  | $A$ | $B$ | $\alpha$ | $\beta$ |
|---|---|---|---|---|
| **Preferred spatial frequency** | 8.1 $CI_{95}$ = [7.4, 8.8] | 2.1 $CI_{95}$ = [1.6, 2.5] | 0.09 $CI_{95}$ = [0.07, 0.12] | 0.02 $CI_{95}$ = [0.00, 0.04] |
| **Cortical magnification** | 17.9 $CI_{95}$ = [17.2, 18.8] | 1.2 $CI_{95}$ = [1.0, 1.4] | 0.20 $CI_{95}$ = [0.18, 0.24] | 0.03 $CI_{95}$ = [0.00, 0.06] |

Increasing $A$ represents a shift upwards in the eccentricity function (Fig 4A). Decreasing $B$ represents a rightwards shift in the eccentricity function (Fig 4A). $\alpha$ and $\beta$ represent the asymmetry between the horizontal and vertical, and lower than upper vertical meridian (Fig 4B), respectively. 95% CIs are derived from bootstrapping across observers.

To assess whether fitting the model to the group-average data introduced distortions, we also fit the 2D model to each individual participant and compared the results. The average parameters fit to each observer separately (S3 Table) were very similar to the parameters fit to the group-average data.

## Polar angle asymmetries

The 2D model included two parameters, $\alpha$ and $\beta$, accounting for polar angle asymmetries. When $\alpha$ is positive, the V1 property is higher along the horizontal than vertical meridian, and when $\beta$ is positive, the V1 property is higher along the lower than upper vertical meridian. To compare to prior studies, we computed a more typical asymmetry index for the two asymmetries. These index values were derived directly from the binned data –without the model fit– using the segment values between 1.5° and 9.5° of eccentricity and ±22.5° of polar angle from the cardinal meridians.

We calculated an HVA index as:

$$HVA = \frac{horizontal\ meridian - vertical\ meridian}{mean(horizontal\ meridian,\ vertical\ meridian)} \times 100$$

Equation 5

An HVA index of 0 indicates the same values for the horizontal and vertical meridians. As the asymmetry between the horizontal and vertical meridians increases, so does the magnitude of the HVA index.

Similarly, we calculated a VMA index as:

$$VMA = \frac{lower\ vertical\ meridian - upper\ vertical\ meridian}{mean(lower\ vertical\ meridian,\ upper\ vertical\ meridian)} \times 100$$

Equation 6

A positive VMA index indicates a higher value on the lower than upper vertical meridian.

For preferred spatial frequency, the median HVA index computed from the data was 31 ($CI_{95}$ = [24, 38]) and the VMA index was 16 ($CI_{95}$ = [4, 25]). For cortical magnification, the asymmetries were larger; the median HVA index computed from the data was 74 ($CI_{95}$ = [64, 81]) and the VMA index was 23 ($CI_{95}$ = [10, 36]). We plot all of these measurements and indices for cortical magnification from four prior studies [14,20,32,33] in Fig 6. For both properties –preferred spatial frequency and cortical magnification– all studies report positive HVA and VMA. Further, within all studies, the HVA index is larger than the VMA index. Note that for consistency and comparison to prior reports, the asymmetry indices for both cortical magnification and preferred spatial frequency are derived from areal measures (mm$^2$/deg$^2$ and cycles$^2$/deg$^2$).

The analyses above quantified preferred spatial frequency from data averaged across stimulus orientation (pinwheels and annuli). However, preferred spatial frequency depends on the local stimulus orientation [11–13] and may interact with polar angle location. In a supplementary analysis we separated out the effects of local orientation and polar angle on preferred spatial frequency (S1 Text: *Three anisotropies in spatial frequency tuning*).

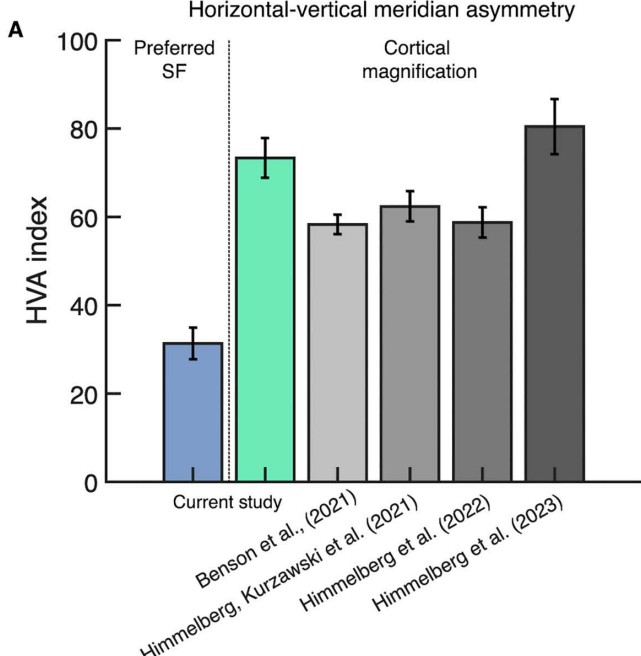
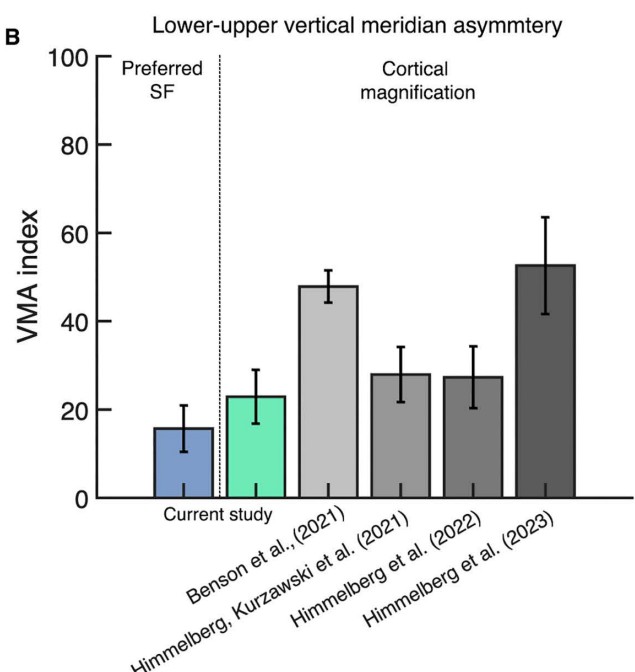

**Fig 6. HVA and VMA indices for preferred spatial frequency and cortical magnification. (A)** Mean HVA indices; the colored plots show the HVA index for preferred spatial frequency and cortical magnification from the current study (n=40). The gray plots show the HVA index for cortical magnification derived from previous reports; Benson et al. (2021) [14]: 1–6° eccentricity, 20° angle, n=163; Himmelberg, Kurzawski et al. (2021) [33]: 1-8° eccentricity, 25° angle, n=44; Himmelberg et al (2022) [20]: 1–8° eccentricity, 15° angle, n=29; Himmelberg et al. (2023) [32]: 1–7° eccentricity, 25° angle, n=24. (B) is the same as (A) but VMA indices. Current study error bars show ±1 SD across bootstraps across participants, and prior study error bars show ±1 standard error of the mean (SEM) across participants.

## Covariation between V1 preferred spatial frequency and cortical magnification

Next, we quantified how preferred spatial frequency and cortical magnification covaried as a function of eccentricity, polar angle, and individual observer. To do so, we summarized both properties –preferred spatial frequency and cortical magnification– separately for each dimension (eccentricity, polar angle, and individual observer) by pooling over the other two dimensions.

We first assessed how preferred spatial frequency and cortical magnification covary with eccentricity. As expected, preferred spatial frequency (Fig 7 A) and cortical magnification (Fig 7 B) declined with eccentricity according to an inverse linear function. The specific way in which they declined was highly similar, which we summarized in two ways. First, we computed their correlation, which was high: $r$=0.99 (CI$_{95}$ = [0.99, 1.00], across 50 bootstrapped group-averages), indicating that these properties tightly covary across eccentricity when pooled across polar angle and observers. Second, we asked whether the two properties are not just correlated but also proportional. We did so by comparing a line of proportionality fit to the data (black dashed line in Fig 7C: $y=mx$) against a line fit to the data with an intercept that was free to vary (green line in Fig 7C: $y=mx+b$). The two lines are very similar (line of proportionality ($m$=0.37; CI$_{95}$ = [0.35, 0.39]) vs line fit to data ($m$=0.32; CI$_{95}$ = [0.29, 0.34]), indicating that the two variables are proportional. The slope of the line of proportionality, $m$=0.37, has interpretable units: at any eccentricity, one cycle of a stimulus at the preferred spatial frequency extends over 2.7 mm (1/$m$) of cortex.

Preferred spatial frequency and cortical magnification also covaried with polar angle, but the pattern of covariation differed from the pattern observed for eccentricity. Preferred spatial frequency (Fig 7D) and cortical magnification (Fig 7E) were modulated by polar angle; both were greater at the horizontal than vertical meridian, and at the lower than upper

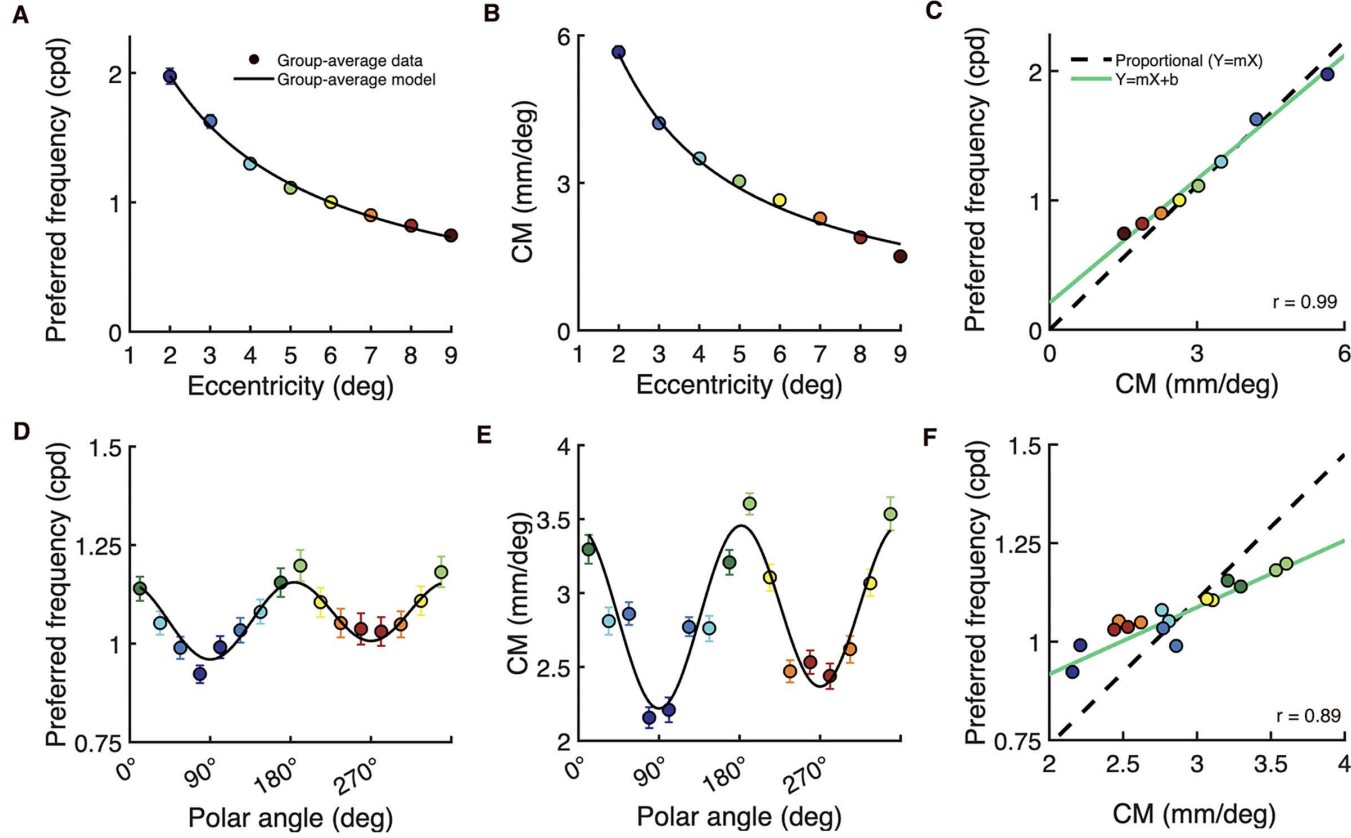

**Fig 7. Covariation between V1 preferred spatial frequency and cortical magnification when measured as a function of eccentricity and polar angle.** Measurements of **(A)** preferred spatial frequency and **(B)** cortical magnification (CM) vary as a function of eccentricity (summarized over polar angle). The colored data points represent group-average measurements and the black line represents the 2D models (summarized over polar angle) fit to the data. **(C)** Covariation between preferred spatial frequency and CM as a function of eccentricity; the colored data points come from (A) and (B). The line of proportionality ($y=mx$) is shown as the black dashed linear fit through the data and an ordinary least products (OLP) regression line ($y=mx+b$) is shown in green. **(D-F)** The same as above, but the preferred spatial frequency and CM measurements –and model fits– vary as a function of polar angle (summarized over eccentricity). Error bars represent ±1 SD cross 50 bootstrapped group-averages.

vertical meridian of the visual field. The correlation between the two properties was high: $r=0.88$; $CI_{95} = [0.76, 0.94]$, indicating that preferred spatial frequency and cortical magnification tightly covary when measured around polar angle (Fig 7F). However, the slope of the line fit with an intercept ($m=0.17$; $CI_{95} = [0.135, 0.20]$) was only half the slope of the line of proportionality ($m=0.37$; $CI_{95} = [0.35, 0.39]$) (Fig 7F). The green line with an intercept is clearly a better fit to the data than the black dashed line of proportionality. This indicates that preferred spatial frequency does not scale with cortical magnification when measured as a function of polar angle. This is consistent with our prior observation that polar angle asymmetries for cortical magnification are about twice as large as those for preferred spatial frequency (see Fig 6).

We can consider these two patterns of covariation through the lens of our 2D models, which are separable in polar angle and eccentricity. The implication is that within any given polar angle, cortical magnification and preferred spatial frequency scale as a function of eccentricity, but the scale factor differs among polar angles. Specifically, the horizontal meridian has the largest ratio of mm per cycle (at the preferred spatial frequency), and the upper vertical meridian has the smallest. According to our 2D model parameter estimates (Table 1), this corresponds to a ratio of 2.7 mm per spatial frequency cycle on the horizontal meridian and 2.0 mm per spatial frequency cycle on the upper vertical meridian.

Finally, we assessed how V1 preferred spatial frequency covaries with cortical magnification when measured as a function of individual observer. To do so, we computed metrics of overall V1 preferred spatial frequency and overall V1 cortical magnification for each individual observer; these 'overall' metrics quantify and summarize preferred spatial frequency or cortical magnification throughout the full V1 representation (1.5–9.5° eccentricity, 0–360° of angle). See methods for details. The coefficient of variation ($\sigma/\mu$) for overall preferred spatial frequency was .35 and the coefficient of variation for overall cortical magnification was 0.15, indicating substantial individual variability in both properties.

The correlation between preferred spatial frequency and cortical magnification was positive and robust, $r = 0.41$; $CI_{95} = [0.16, 0.64]$, indicating that preferred spatial frequency covaries with cortical magnification at the individual level (Fig 8). However, this correlation was lower than the correlations as a function of eccentricity ($r = 0.99$) and polar angle ($r = 0.88$). The general pattern is that observers with greater overall V1 cortical magnification (i.e., a larger V1) tend to have a higher overall V1 preferred spatial frequency, and vice versa. The data, as summarized by the linear fit ($m = 0.99$; $CI_{95} = [0.69, 1.40]$), are steeper than the line of proportionality ($m = 0.40$; $CI_{95} = [0.38, 0.41]$). This reflects greater variation across individuals for overall preferred spatial frequency than overall cortical magnification.

The correlation across individuals was computed using overall summary metrics for preferred spatial frequency and cortical magnification. Because these metrics integrate across the visual field, they have relatively high signal to noise. In a separate analysis, we computed 2D models of preferred spatial frequency and cortical magnification for each observer, enabling us to assess the relation between preferred spatial frequency and cortical magnification in a more granular way. We tested whether observers with a larger asymmetry for cortical magnification have a larger asymmetry for preferred spatial frequency. To do so, we correlated the $\alpha$ and $\beta$ parameters of the 2D models when fit to preferred spatial frequency and cortical magnification. Neither parameter was correlated across observers (HVA: $r = -0.10$; $CI_{95} = [-0.50, 0.31]$, VMA: $r = -0.24$; $CI_{95} = [-0.64, 0.29]$), indicating that, on the individual level, the extent of either asymmetry for V1 cortical magnification does not correlate with asymmetry for preferred spatial frequency.

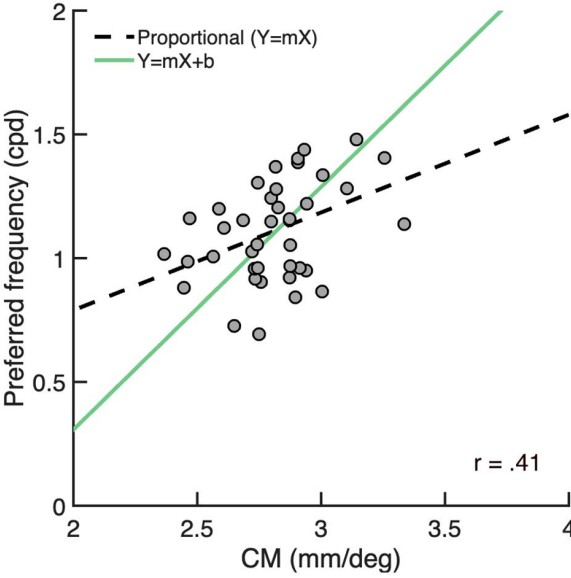

**Fig 8. Covariation between overall V1 preferred spatial frequency and overall V1 cortical magnification as a function of individual observer.** Each gray data point represents an individual (n = 40). The line of proportionality ($y = mx$) is shown as the black dashed linear fit through the data and an ordinary least products (OLP) regression line ($y = mx + b$) is shown in green.

## Discussion

We quantified V1 preferred spatial frequency and cortical magnification as a function of eccentricity, polar angle, and individual observer. There were several parallels between preferred spatial frequency and cortical magnification. First, both declined with eccentricity according to an inverse linear function. Second, both properties showed polar angle asymmetries: they were larger along the horizontal than vertical, and along the lower than upper vertical meridian representation of the visual field. Third, the two properties covaried among individual observers: a V1 with greater overall cortical magnification was tuned to higher spatial frequencies. Despite these patterns of covariation, there was a striking difference comparing the two properties as a function of different visual field dimensions: As a function of eccentricity, the change in cortical magnification and preferred spatial frequency were proportional, but as a function of polar angle, they were not. When measured as a function of polar angle, cortical magnification varied more than twice as much as preferred spatial frequency, indicating larger polar angle asymmetries for cortical magnification than for preferred spatial frequency. The opposite pattern was found across individuals: preferred spatial frequency varied more than cortical magnification. We discuss implications of these patterns below.

### Relation to other models of V1 topography

One of the most striking features of V1 is its inhomogeneity. For example, about half of V1 represents the central 12° of the visual field –less than 2% of visual space; according to the Horton and Hoyt [7] estimate, in which Magnification $=300/$ (Eccentricity+0.75)$^2$, a full visual hemifield (out to 90°) projects to $3600\,mm^2$. A hemifield out to 12° radius projects to $1800\,mm^2$, i.e., half the surface area. In contrast, in the visual field, a hemfield with a 90° radius has 50 times more surface area than one with a 12° radius Polar angle asymmetries in the organization of V1 are also large, but have received much less attention than eccentricity effects: cortical magnification is 2x higher along the horizontal meridian than vertical meridian, and about 1.4x higher along the lower than upper vertical meridian (Fig 6) [14,15,20,32,33].

Most models of V1 topography assume symmetric properties as a function of polar angle (for review see [30]). However, the V1 models that do have polar angle asymmetries were not fit to cortical measurements of the HVA and VMA. The widely used model from Rovamo and Virsu [29] was derived prior to the advent of functional neuroimaging. Their estimates thus came from human retinal ganglion cell (RGC) data and phosphene mapping, assuming that V1 uniformly samples the retina. However, with retinotopic mapping using fMRI, it became possible to measure cortical magnification precisely and show that V1 does not uniformly sample the RGCs. Rather, polar angle asymmetries in RGC density are amplified in cortex [34,35]. Next, the 'log-polar' mapping proposed by Schwartz [8,9] describes an inverse HVA (greater cortical magnification for the vertical than horizontal meridian) and no VMA. The greater cortical magnification for the vertical than horizontal meridian in the Schwartz model [8,9] and its subsequent variants [36] are an incidental result of a design constraint (equal magnification in the radial and tangential directions) rather than being informed by cortical measurements. A new variant of the log-polar mapping was proposed to intentionally eliminate polar angle asymmetries as an unwanted feature of the Schwartz model [37]. Finally, a series of studies proposed a model of V1 and extrastriate topography that is defined on an 'average' cortical surface template [38–40]. Unlike the log-polar models, these surface-based models do not have a simple parametric description due to the complexity of the folding pattern on the cortical surface and its variability from person to person.

Hence there has been no model of human V1 topography in a simple parametric form that incorporates the known polar angle asymmetries. Our 2D model of V1 extends the inverse-linear equation for cortical magnification as a function of eccentricity [7], providing a simple parametric form and incorporates both eccentricity and polar angle.

### Scaling in the human visual system

Our 2D models enabled us to make quantitative comparisons between different V1 properties. These comparisons showed that preferred spatial frequency and cortical magnification are proportional as a function of eccentricity. The

proportionality suggests a common cause for the two functions. One possibility is that both properties depend on the density of RGCs. RGC density declines inverse linearly with eccentricity [41,42], similar to our measurements of preferred spatial frequency and cortical magnification. Using the formula of Watson [42] for RGC density, we computed two functions: the number of RGC inputs for a patch of V1 representing 1 cycle of a stimulus at the preferred spatial frequency, and the number of RGC inputs per mm of V1 surface. These functions are relatively flat across the eccentricity range we studied (2–9°): about 45 RGCs per cycle and 18 RGC per mm (Fig 9A, and 9B, black dashed line). By comparison, both preferred spatial frequency and cortical magnification measured with respect to the visual field vary sharply with eccentricity (Fig 9C and 9D, black dashed line). Therefore, we speculate that most of the decline in preferred spatial frequency and cortical magnification as a function of eccentricity is inherited from the retina.

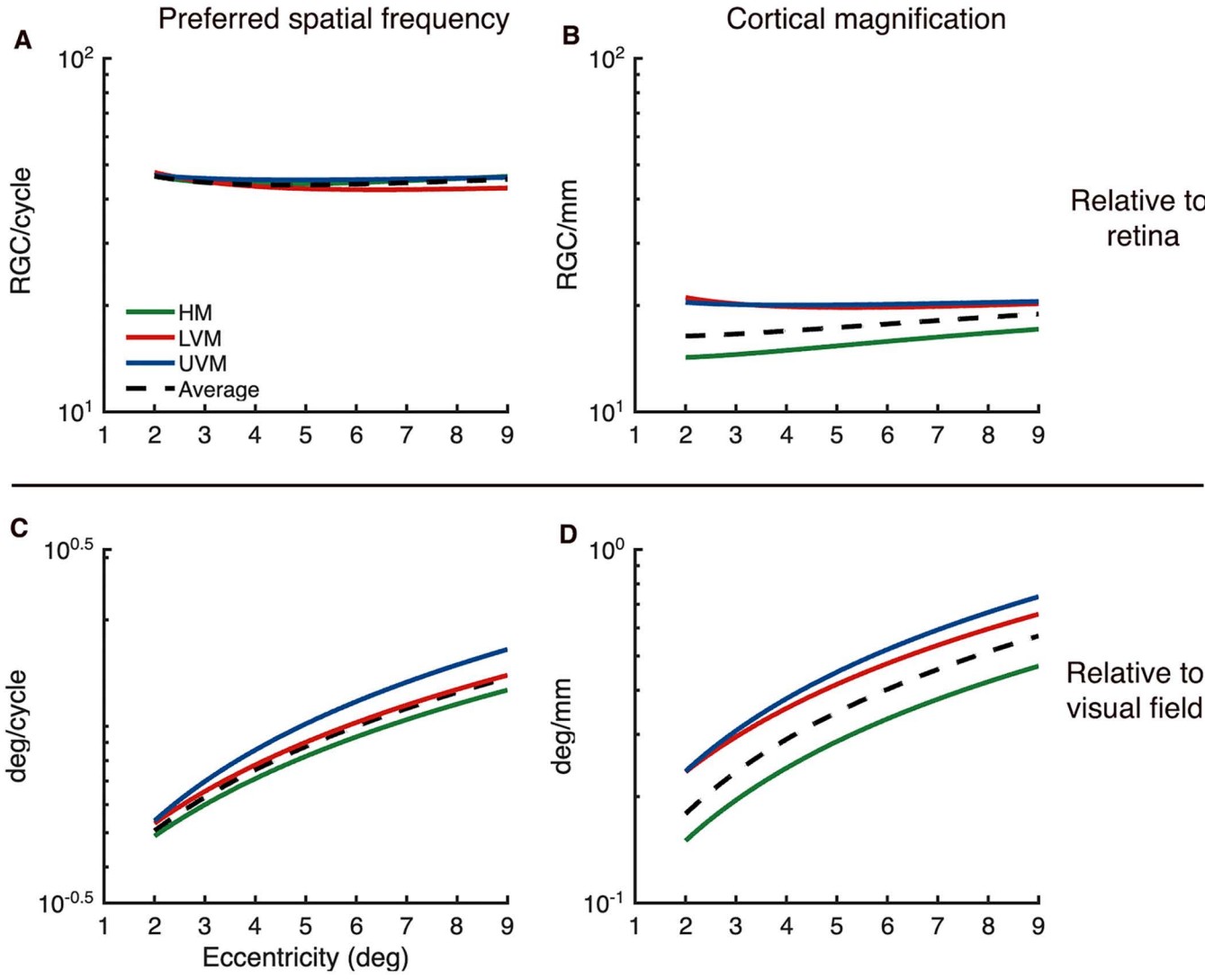

**Fig 9. Calculations of retinal ganglion cell (RGC) per spatial frequency cycle and per mm of V1 cortex. (A)** Measurements of RGCs per cycle of stimulus spatial frequency at preferred spatial frequency as a function of eccentricity, for the 3 meridians and the average of V1. **(B)** The same as (A) but for RGCs per mm of V1 surface area. **(C)** Measurements of degrees of visual space per cycle of stimulus spatial frequency at preferred spatial frequency as a function of eccentricity. **(D)** The same as (C) but for degrees of visual space per mm of V1 surface area.

The fact that these two properties are proportional as a function of eccentricity is in agreement with the hypothesis that the V1 map consists of repeated, identical units, as previously proposed [43,44]. Notably, certain features of V1—such as the width of ocular dominance columns and the rate at which preferred orientation changes along the cortex—remain constant across eccentricity [43,44]. These constancies do not contradict the fact that cortical magnification [7] and preferred spatial frequency [11–13] vary across the V1 map. Instead, according to the hypothesis of cortical constancies, these declines arise from changes in inputs to cortex as a function of eccentricity, rather than from alterations in cortical circuitry.

This hypothesis of cortical constancies implies a proportional relation between cortical magnification and preferred spatial frequency for the following reasons. Both properties are expressed as ratios relative to the visual field (mm/deg and cycles/deg). When their ratio is computed, the denominators –degrees of visual field– cancel out. The numerators –mm of cortex and spatial cycles– depend on how many V1 cells process a given number of input fibers and how finely these cells compare the inputs, respectively. The *constant circuit hypothesis* suggests that these values are constants across the cortex, leading to a proportional relationship between the two properties.

In contrast to eccentricity, the constancy hypothesis does not hold for the polar angle dimension. Relative to RGC density, both preferred spatial frequency and cortical magnification are flat along the eccentricity dimension (Fig 9A, and 9 B black lines). Relative to RGC density, preferred spatial frequency is also relatively constant as a function of polar angle, indicated by the tight clustering of the polar angle lines in Fig 9A. For all polar angles, we estimate about 45 RGCs per cycle at the preferred spatial frequency. In contrast, relative to RGC density, cortical magnification does vary substantially with polar angle, indicated by the spread of the polar angle lines in Fig B. For example, for the horizontal meridian, we estimate about 15 RGCs per mm of cortex, whereas for the upper vertical, we estimate about 20 RGCs per mm of cortex.

This pattern clarifies the interpretation of our covariation results. We reported that, as a function of polar angle, preferred spatial frequency and cortical magnification covaried but were not proportional. The comparison to RGC density shows that the polar angle asymmetries in cortical magnification *amplify* the retinal pattern, as noted previously [34]. In contrast, the polar angle asymmetries in preferred spatial frequency approximately *track* the retinal pattern (Fig 9A). This means that variation in V1 properties along the eccentricity axis can be explained by differences already present in the retina; however, polar angle asymmetries arise, at least in part, from post-retinal mechanisms, either in the LGN or cortex. It is not known why the cortex amplifies the polar angle asymmetries in cortical magnification, but there are many cortex-specific properties of V1 that vary with polar angle. For example, sulcal patterns and wiring tension differ between the horizontal and vertical meridian representations [45]. Interhemispheric connectivity through the splenium is present along the vertical but not the horizontal meridian [46]. Further, the length of connections between the horizontal meridians of V1 and V2 are much longer than those connecting vertical meridians, as the horizontal representations are much further apart on the cortex. These, and other sources of variation in cortex-specific structural properties around polar angle, might affect cortical surface area, and thus cortical magnification, thereby giving rise to differences in polar angle asymmetries between cortex and retina.

### Larger V1s prefer higher spatial frequencies

Complementing the finding that the two V1 properties covary with eccentricity and polar angle, we also found that overall V1 preferred spatial frequency and cortical magnification were correlated among individual observers. Assuming equal neural density across observers, a larger V1 has more neurons. Greater neural count likely allows for finer sampling of visual space and thus higher preferred spatial frequency. This is supported by the observations that V1 surface area is inversely correlated with receptive field size [24,47,48] and that V1 neurons with small receptive fields are typically tuned to higher spatial frequencies [6,49–51].

Although the correlation between V1 size and preferred spatial frequency was positive and significant across observers, the strength of the correlation was lower ($r = 0.36$) than across polar angle ($r = 0.89$) or eccentricity ($r = 0.99$). Several factors likely contribute to the lower correlation across individuals. First, the assumption of equal neural density across

individuals is not exact. V1 neural density varies more across pairs of individuals than between sites within an individual V1, as estimated from postmortem data [52,53]. Hence, cortical magnification is a better proxy for neural count when comparing across locations within a V1 map than between individual V1 maps. Additionally, there are analysis factors that limit the strength of the correlation across individuals. There is less averaging, and therefore lower SNR, for the individual difference analysis than the location analyses. The data were binned into 8 eccentricities $\times$ 16 polar angles $\times$ 40 individuals. To compute covariation across one dimension, we collapsed (i.e., averaged) across the other two. Therefore, we averaged across the most bins for eccentricity (16 $\times$ 40) and the fewest for individual differences (8 $\times$ 16). The decreasing amount of averaging parallels the declining correlation coefficients. Finally, the dynamic range differs across the three dimensions. Cortical magnification increased more than 200% across eccentricity (from 9.5° to 1.5°), about 70% across polar angle (from the upper vertical to horizontal meridian), and about 50% across individuals (from the smallest to largest V1), also paralleling the declining correlation coefficients.

**Implications for visual perception**

Our two dependent variables were cortical magnification and preferred spatial frequency. Here we consider the implications of variability in each property for visual perception.

Cortical magnification declines with eccentricity; thus any visual task that declines with eccentricity will be correlated with cortical magnification. Here, we focus on contrast sensitivity because it has been tightly linked to cortical magnification both quantitatively and mechanistically. When stimulus size is held constant, contrast sensitivity declines with eccentricity [54–56]. Virsu and Rovamo [57] found that this decline could be explained by the decreasing area of cortex stimulated at greater eccentricities due to the decline in cortical magnification. Their interpretation was that the critical factor governing detection was the number of neurons stimulated. To test this, they showed that the decline in contrast sensitivity could be eliminated by scaling the stimulus size inversely with magnification (M-scaling) to equate the size of the stimulated cortex, thereby establishing a tight link between cortical magnification and contrast sensitivity.

The same logic can be extended to polar angle. Both cortical magnification [14,15,20,32,33] and contrast sensitivity [20,54,58–65] are greater along the horizontal than vertical meridian, and along the lower vertical than upper vertical meridian (see [66] for review). Moreover, the extent of the asymmetries are about 2–3 times larger for the HVA than the VMA for both cortical magnification (Fig 7) [14,20,32,33] and contrast sensitivity [60,63]. However, scaling stimulus size inversely with cortical magnification along the polar angle meridians, using the equation set forth by Virsu and Rovamno [57], does not eliminate differences in contrast sensitivity as a function of polar angle [63]. Because their magnification functions are estimated from RGC measurements [57] –rather than detailed cortical magnification measurements– scaling stimulus size according to more recent V1 cortical magnification data might eliminate polar angle differences. Finally, at the individual level, a larger V1 means greater overall cortical magnification and, presumably, a greater neural count. This would suggest that a larger V1 should be associated with greater contrast sensitivity [29], as we previously found when measuring contrast sensitivity with an orientation discrimination task [20]. The specific behavioral measure is important, as, for example, overall V1 surface area was reported to be uncorrelated with performance on a contrast discrimination task [67].

What are the implications of our measurement of V1 preferred spatial frequency for visual perception? The most natural comparison would be to perceptual spatial frequency tuning, for example as measured by the contrast sensitivity function (CSF). Indeed, the spatial frequency at which the CSF peaks declines with eccentricity [57,68–70], paralleling our V1 preferred spatial frequency measurements.

However, it is not clear whether the spatial frequency at which the CSF peaks varies with polar angle: it has been reported to be either asymmetric [62] or similar [61,63] among meridians. This is different from our V1 preferred spatial frequency measurements, which vary strongly with polar angle. This difference may be due to stimulus choice: fMRI experiments use high contrast stimuli to drive large BOLD responses, whereas behavioral contrast sensitivity experiments

use low contrast stimuli to measure thresholds. In fact, behavioral measurements of spatial frequency using high contrast stimuli, including spatial frequency discrimination and perceived spatial frequency [71], and spatial frequency cut-off [72], vary with polar angle, in parallel with what we find in cortical tuning. It is likely that quantitative links between fMRI and psychophysical measurements of spatial frequency tuning can be clarified by closely matching the stimuli and making both sets of measurements in the same observers.

## Conclusion

We measured how preferred spatial frequency and cortical magnification vary throughout the V1 visual field representation: both decline with eccentricity following an inverse linear function and both are greater along the horizontal than vertical, and the lower than upper vertical meridian. Although the two properties were highly correlated across both visual field dimensions, they were proportional as a function of eccentricity, but not polar angle. This translates to a cortical constancy (mm per cycle of visual stimulus) for eccentricity but not for polar angle. The high correlations along both visual field dimensions, as well as the correlation across observers, indicate a tight link between the neural properties. The constancy as a function of eccentricity but not polar angle indicates an important distinction in the organization of V1 for these visual axes.

## Methods

### Ethics

All observers provided written informed consent and consented to the public release of their anonymized data. The experiment was conducted in accordance with the Declaration of Helsinki and was approved by the New York University ethics committee on activities involving human observers (IRB protocol number IRB-FY2022–6427).

### Participants

40 observers (26 females, 14 males, mean age = 30 years, including two authors: M.M.H and Y.K.) were recruited from New York University. All observers had normal or corrected-to-normal vision (contact lenses or MRI-compatible goggles) and completed two scan sessions: a 1–1.5 hour session to measure retinotopic maps and cortical anatomy and a 1 hour session to measure spatial frequency tuning.

### fMRI stimulus display

Observers viewed stimuli from inside the MRI scanner bore using a ProPixx DLP LED Projector (VPixx Technologies Inc., Saint-Bruno-de-Montarville, QC, Canada). The stimulus was projected onto an acrylic back-projection screen (60 cm x 36.2 cm) inside the scanner bore. Observers viewed the screen at a distance of about 83.5 cm (from eyes to the screen) using an angled mirror mounted onto the head coil. The projected image had a resolution of 1920 x 1080 and a refresh rate of 60Hz. The display was calibrated using a linearized lookup table and the display target luminance was 250 cd/m$^2$.

### Retinotopic mapping stimuli and experimental design

Retinotopic maps were measured using population receptive field (pRF) mapping [21] from which we derived eccentricity and polar angle estimates throughout visual cortex. The pRF stimulus was generated on an iMac computer using Matlab 2017a and was projected onto the fMRI stimulus display within the scanner bore using the Psychophysics Toolbox v3 [71] and customized *vistadisp* software (https://github.com/WinawerLab/vistadisp, adapted from https://github.com/vistalab/vistadisp).

Of the 40 observers, 20 completed a pRF mapping experiment that used a drifting bar stimulus for all scans, and 20 completed a pRF mapping experiment that used a bar stimulus for half the scans and a combination of expanding/

contracting wedges and rotating rings for the other half. All other stimulus and scan parameters were identical between these two pRF experiments, including the carrier images. We first describe the experiment using the drifting bar stimulus. The drifting bar pRF stimulus consisted of a carrier (i.e., image patterns) windowed by a bar aperture that swept across the screen during each scan. The experiment was identical to that used in [33]. In brief, the bar apertures were limited by a circle of 12.4 degrees eccentricity and drifted across the screen 8 times in different directions, 24 s per sweep. The carrier contained colorful objects, faces, and scenes at multiple scales [72] superimposed on an achromatic pink noise (1/f) background. Each scan lasted 192 s, and observers completed between 6 and 10 scans.

In the second version of the retinotopy experiment we used drifting bar apertures for half of the scans, and a combination of expanding/contracting ring and rotating wedge apertures for the other half. This design was adapted from the NSD dataset [73]. The apertures were sweeping bars for 3 scans, and a combination of rotating wedges, and expanding and contracting rings for the other 3 scans, with the bars and "wedgering" scans interleaved. The aperture patterns were the same as those in the bar-only scans, as was the carrier update frequency, aperture step frequency, and spatial extent. For the rotating wedge component, the circular aperture was revealed by a 90° wedge that swept around the screen clockwise or counter-clockwise in 8 steps. For the ring component, the circular aperture was revealed by a ring that expanded then contracted across the circular aperture in 8 steps. The size of the ring linearly changed with eccentricity. Each scan was 300 s in length, and observers completed 6 scans.

Inspection of the data revealed that the retinotopy experiment – bars only versus bars/wedges/rings – resulted in no differences in cortical magnification as a function of eccentricity.

## Spatial frequency stimuli

The stimuli used in the spatial frequency experiment were adapted from [11]. The stimuli consisted of 20 log-polar gratings, such that the local spatial period (the reciprocal of spatial frequency) was proportional to eccentricity and invariant to polar angle. There were 2 stimulus classes: pinwheels and annuli, defined by Equation 7:

$$I(r, \theta) = cos(\omega_r ln(r) + \omega_a \theta + \phi)$$

Equation 7

where $I$ is the local pixel intensity, $\omega_r$ is the radial component of the base frequency, $\omega_a$ is the circular component of the base frequency, $r$ is the eccentricity in deg, $\theta$ is the polar angle in radians, and $\phi$ is the phase (in radians). For pinwheels, $\omega_r$ is 0 and $\omega_a$ is the base frequency, and *vice versa* for annuli. There were 10 spatial scales for each stimulus class with base frequencies of 3, 5, 8, 13, 21, 35, 58, 95, 156, and 256. For pinwheels, the base frequency has units of cycles/revolution. For annuli, the base frequency has units of cycles/log deg. For a given base frequency, the local spatial frequency (in cycles/deg) is identical for the annulus and pinwheel at all corresponding points in the image. Note that our stimulus range, 3 to 256, is larger than that in [11] (6 to 128) because the authors found that V1 spatial frequency tuning in a voxel was wide, about 5 octaves measured as the full-width at half-max of a log-Gaussian tuning function.

The stimuli were masked by a circular aperture (12° radius). A uniform gray anti-aliasing mask covered the center of the stimuli (1.2° radius, except for the stimuli with a base frequency of 256 for which the anti-aliasing mask was 1.5° radius). Thus, all analyses were limited to vertices with a pRF center eccentricity greater than 1.5°.

Because of optical imperfections in the projected images, the rendered spatial frequency was not identical to the spatial frequency in the image files. To find the rendered spatial frequency, we inverted the modulation transfer function of the projector, as described previously [11]; https://github.com/WinawerLab/spatial-calibration).

## Spatial frequency: experimental design

The spatial frequency stimuli were presented by an iMac computer, running Psychophysics Toolbox v3 [71] with Matlab 2017a. Stimuli were projected onto the display in the fMRI scanner bore. Each scan began with 16 s of a fixation cross

on a uniform gray screen. Each stimulus trial consisted of a 4 s presentation of one stimulus condition. During each trial, stimuli were presented in an on-off fashion 8 times (250 ms on, 250 ms off). The phases of the 8 stimuli varied to minimize neural adaptation. Each of the 20 stimulus conditions (pinwheels and annuli with 10 base frequencies) were presented 3 times per scan. The order of stimulus conditions was randomized within each scan and each of the 8 different phases was random within each trial. In addition to the 60 stimulus trials per scan, there were 6 blank trials (4 s gray screen) randomly intermixed with the stimulus trials. At the center of the display, within the anti-aliasing mask, there was a fixation cross that changed color from red to green and *vice versa*. Observers were instructed to maintain fixation on this cross throughout the experiment and respond, via the button box, when it changed color. Fixation was live-monitored using an EyeLink 1000 eye tracker [74] within the scanner. Each scan ended with 16 s of blank display, except for the fixation cross. Each observer completed between 6 and 8 scans, which lasted 296 s each, except for one observer who completed 4 scans. The number of scans was limited by how long it took to complete the one-hour session.

## Acquisition of anatomical and functional data

Acquisition was the same as our previous study [33], summarized here for the reader's convenience. Anatomical and functional data were acquired on a 3T Siemens MAGNETOM Prisma MRI scanner using a Siemens 64-channel head coil. A T1-weighted (T1w) MPRAGE anatomical image was acquired for each observer during the retinotopy scan session (TR, 2400 ms; TE 2.4 ms; voxel size, 0.8 mm$^3$ isotropic; flip angle, 8°). The anatomical image was auto-aligned to a template to ensure similar slice prescription for all observers. Functional echo-planar images (EPIs) were acquired for each observer using a T2-weighted multiband EPI sequence (TR, 1000 ms; TR, 37 ms; voxel size, 2mm$^3$, flip angle, 68°; multiband acceleration factor, 6; phase-encoding, posterior-anterior) [75,76]. Two distortion maps were also acquired to correct susceptibility distortions in the functional images; one spin-echo image with anterior-posterior (AP) phase encoding and one with posterior-anterior (PA) phase encoding.

## Processing of anatomical data

fMRIprep v. 20.0.1 [77,78] was used to preprocess the anatomical scan. For each observer, the T1w anatomical image was corrected for intensity inhomogeneity and then skull stripped. The anatomical image was automatically segmented into cerebrospinal fluid, white matter, and gray matter using *fast* [79]. Cortical surfaces were reconstructed using Freesurfer's recon-all [80] and an estimated brain mask was refined using a custom variation of the method.

## Processing of functional data

All functional data were processed as follows. First, a reference volume –and skull stripped version– was generated using the custom methodology of *fMRIPrep*. The AP and PA distortion maps were used to estimate a B0-nonuniformity map. The estimated distortion of the B0-nonuniformity map was then used to generate a corrected functional reference image, which was then coregistered to the anatomical image using six degrees of freedom. Head motion parameters –with respect to the functional reference– were then estimated before any spatiotemporal filtering. Each EPI was slice-time corrected, and all slices were realigned to the middle of each TR. The slice-time corrected functional data were then resampled to the T1w anatomical space via a one-shot interpolation consisting of all the pertinent transformations (i.e., head motion transform matrices, susceptibility distortion correction). The preprocessed time-series data were then resampled to the *fsnative* surface by averaging across the cortical ribbon. All further analyses were conducted on the *fsnative* surface in vertex space for individual observers.

## Implementing the pRF model

The pRF model was implemented on each observer's *fsnative* surface as derived from *Freesurfer*. For each vertex, the time-series data across the functional pRF scans were averaged together to generate an average time-series. These

average time-series were then transformed to BOLD percent signal change (i.e., % change at each TR from the mean signal across all TRs). The pRF model was fit to the BOLD signal change at each vertex. The pRF model was implemented using *vistasoft* (https://vistalab.stanford.edu/software/, Vista Lab, Stanford University). We used customized Matlab wrapper code to run the pRF model on the *fsnative* surface (https://github.com/WinawerLab/prfVista). The pRFs were modeled as a circular 2D-Gaussian parameterized by *x*, *y*, and σ. The *x* and *y* parameters specify the center position of the 2D-Gaussian in the visual field, and the σ parameter, the standard deviation of the 2D-Gaussian, specifies the size of the receptive field. The 2D-Gaussian was multiplied pointwise by the stimulus contrast aperture and convolved with a hemodynamic response function (HRF) to predict the BOLD percent signal change. We parameterized the HRF used in the pRF model by five values, describing a difference of two gamma functions [81,82].

The pRF model was implemented using a multistage coarse-to-fine approach [24,32] that reduces the chance of the search algorithm getting stuck in a local rather than global optimal solution and reduces the chance of finding a solution that is noise rather than signal. The data were first temporally downsampled and smoothed (decimation) by a factor of two to remove high frequency noise. Next, the *x*, *y*, and σ parameters were fit using a brute force grid search. These results were taken as the starting point of a second-stage search fit. The estimated parameters were held fixed and the HRF parameters were then fitted by a search that chose parameters minimizing the squared error between the data and the prediction averaged across vertices. In the final stage of the approach, the HRF parameters were held fixed and the pRF parameters were refitted to the data in a search fit. These final *x* and *y* values were then used to compute vertex-wise eccentricity and polar angle coordinates, reflecting pRF center positions in the visual field. Only pRF data with an $R^2 \geq 10\%$ were included in our analyses.

### Defining V1 and cleaning the V1 maps

V1 was defined as a region-of-interest (ROI) by hand using *Neuropythy* (https://github.com/noahbenson/neuropythy) [40]. The V1 ROI extended from 0° to 9.5° of eccentricity, with the V1/V2 dorsal border falling through the center of the lower vertical meridian, and the V1/V2 ventral border falling through the center of the upper vertical meridian. After defining V1, each observer's polar angle and eccentricity maps were cleaned using Neuropythy [40]; this involves the implementation of an optimization algorithm on the pRF eccentricity and polar angle fits within the V1 ROI. This minimization aims to adjust the pRF centers of the vertices as little as possible to simultaneously enforce a smooth retinotopic map and correct the field sign values throughout V1.

### Estimating spatial frequency response amplitudes using GLMdenoise

For each observer, and each vertex on the *fsnative* surface, we used the GLMdenoise Matlab toolbox [22] to estimate the beta weights for each of the 21 stimulus conditions (20 spatial frequency stimuli and 1 blank condition). The GLMdenoise algorithm generates and fits an observer specific HRF and estimates beta weights for each vertex –and each stimulus condition– with 100 bootstraps across scans. For each vertex, and each stimulus condition, we used the median bootstrapped beta weight in further analyses. The GLMdenoise algorithm also includes 3 polynomial regressors that capture the mean signal and slow drift, and noise regressors derived from vertices that are not well fit by the GLM. The beta weights for each vertex were L2 normalized, to equalize the contributions of different vertices when averaging data. For each vertex, the beta weight for the null stimulus (i.e., mean luminance) was subtracted from each of the 20 beta weights.

### Parcellating V1 into segments

We parcellated each observer's V1 into 128 segments (64 segments per hemisphere). The 128 segments were derived from the intersection of 16 polar angle bins (22.5°-wide bins starting from 0° of polar angle) and 8 eccentricity bins (1°-wide bins from 1.5°–9.5° eccentricity). Each segment was derived by identifying the V1 vertices that had pRF polar angle and eccentricity coordinates that fell within the criterion for a segment, subject to the constraint that the GLM $R^2 \geq 2\%$ for

a vertex. Note that because the GLM variance explained is computed from cross-validated data, the expected value from noise is negative.

Given 128 segments for each of 40 observers, our goal was to compute cortical magnification (mm/deg) and preferred spatial frequency (cycles/deg) within each of these 5120 segments, and to then summarize the data across eccentricity, polar angle, or individual observer. This was feasible for the cortical magnification measure, which just entails summing the surface area of vertices on the cortical surface within each segment of each observer. This was not feasible for preferred spatial frequency, for which we fit log-Gaussian tuning functions to BOLD responses, because this function is a non-linear fit to the data, and can yield implausible estimates when the SNR is low. Hence, as we describe below, when estimating spatial frequency tuning functions, we averaged GLM beta weights before fitting the tuning functions, either across observers when analyzing effects of eccentricity or polar angle, or across V1 polar angle bins when analyzing the effect of individual differences.

## Computing preferred spatial frequency for V1 segments

For each V1 segment in each observer, we computed the mean beta weights across vertices separately for each of the 20 stimulus conditions. Because the stimulus spatial frequency declines with eccentricity, the local spatial frequency for each vertex depends on its pRF location. We relate the local spatial frequency to the base frequency by Equation 8:

$$SF_l = \frac{SF_b}{2\pi r}$$

<div align="right">Equation 8</div>

where $SF_l$ is the stimulus spatial frequency at a vertex's preferred eccentricity, $SF_b$ is the base frequency of the stimulus (cycles per revolution for radial patterns, cycles per log eccentricity for annuli), and $r$ is the eccentricity of the vertex in deg. We then averaged the local spatial frequency, $SF_l$, across vertices within a V1 segment. We did this separately for each of the 20 stimuli. Thus, for each V1 segment, we obtained 20 beta weights and 20 corresponding spatial frequencies.

For analyses of spatial frequency tuning as a function of eccentricity and polar angle, we averaged the beta weights across observers by bootstrap. Specifically, we sampled the 40 observers 40 times with replacement then computed the average and repeated this 50 times. For each bootstrap of each segment, we fit a log-Gaussian tuning function (Equation 9) to 10 beta weights for the pinwheel, the annulus, or the average of the two classes:

$$R = G \cdot \exp\left(-\frac{1}{2}\left(\frac{\log_2 \frac{f}{f_0}}{\sigma}\right)^2\right)$$

<div align="right">Equation 9</div>

where $R$ is the vector of predicted responses (in percent signal change); $G$ is a scale factor (gain); $f$ is the vector of local spatial frequencies; $f_0$ is the preferred spatial frequency, i.e., the spatial frequency at the peak of the Gaussian; and $\sigma$ is the standard deviation in $\log_2$ units (octaves). The fitted parameters are $G$, $f_0$, and $\sigma$. These parameters are estimated by minimizing the squared error between the set of 10 predicted BOLD responses, $R$, and the corresponding 10 beta weights. This log-Gaussian tuning function was fitted independently 6400 times (128 V1 segments x 50 bootstraps).

## Computing cortical magnification for V1 segments

For each observer, cortical surface area maps were generated for each observer using *Freesurfer*. These surface area maps specify the surface area in mm² of each vertex on the midgray representation of the *fsnative* surface. The midgray surface is equidistant from the gray/pial surface and the gray/white surface. We use the midgray to estimate surface area to reduce curvature biases as the gray/white surface is expanded in sulci relative to gyri, and the gray/pial surface is the

opposite. We computed cortical magnification for each V1 segment by dividing the summed surface area (mm$^2$) across vertices within a segment by the area of visual space subtended by the segment (i.e., the solid angle in degrees$^2$). Each segment subtended 22.5° of polar angle, and 1° of eccentricity. Hence, each segment's area is 1/16 of the area of an annulus that is 1° in width, centered at the appropriate eccentricity.

Cortical magnification (mm$^2$ surface area/deg$^2$ of visual space) was then computed for each V1 segment in each observer. We then took the square root of this value to compute linear cortical magnification (mm/deg). We computed the cross-observer average magnification per segment by bootstrapping 50 times across observers. As with preferred spatial frequency, we derived 6400 magnification values (128 segments x 50 bootstraps).

## Summarizing cortical magnification and preferred spatial frequency across a single visual field dimension

For some analyses, we collapsed data across one spatial dimension to visualize only the effect of the other dimension. Doing so requires a series of operations to ensure that the units are correct and interpretable. For one segment of V1, we define the cortical magnification as the cortical surface area (mm$^2$) divided by the visual field area (deg$^2$), and the linear magnification as the square root of this value. When two or more segments are combined, we define the areal cortical magnification of the combined region as the combined surface area divided by the combined visual field area. Simply averaging the magnification factor (linear or areal) of two segments will not give the correct combined magnification if the visual areas of the segments are not identical. Moreover, all calculations for combined regions need to be areal rather than linear to get the correct combined values. The same logic holds for preferred spatial frequency.

Hence, to summarize cortical magnification as a function of eccentricity, we summed the surface areas (mm$^2$) and visual field areas (deg$^2$) of segments across polar angles at each eccentricity and computed the ratio of these sums (mm$^2$/deg$^2$). To convert to linear magnification, we took the square root of this value. To summarize magnification as a function of polar angle, we did the complement: summed the surface areas and visual field areas of segments across eccentricity for each polar angle wedge and computed the ratios.

For preferred spatial frequency, we did the analogous calculations. For preferred spatial frequency, the analog of surface area (deg$^2$) is cycles$^2$. Thus, for each segment, we computed the number of cycles represented by a 2D spatial pattern (plaid) at the preferred frequency by multiplying the square of the preferred spatial frequency (cycles/deg$^2$) by the visual field size of that segment (deg$^2$). To combine preferred spatial frequency across segments, we then summed the number of cycles and divided by the summed visual area, analogous to the magnification calculation. Finally, to convert this to the more traditional linear value (cycles/deg), we took the square root.

## Computing overall V1 cortical magnification and overall V1 spatial frequency preference for individual observers

To correlate dimensions such as preferred spatial frequency vs cortical magnification across observers, it is helpful to reduce each measure to a single value per observer. Cortical magnification, integrated across the visual field, is simply the surface area. To report the individual differences results in the same units as in the visual field analyses, we divided the total surface area by the total visual area included in our analyses (1.5° to 9.5° eccentricity, 360° polar angle), and then calculated the square root, giving us a measure in mm/deg for each observer –overall cortical magnification.

We did the analogous calculation for preferred spatial frequency, but to achieve better signal to noise for estimating tuning functions, for each observer we averaged the beta weights within an eccentricity band (i.e., across polar angles). We then fit the log-Gaussian tuning curve for each eccentricity band for each observer. From the peak of the spatial frequency tuning curve, we computed the number of cycles per eccentricity band, analogous to the surface area of a band for the cortical magnification calculation above. Plots of preferred spatial frequency as a function of eccentricity for individual observers are available in S5 Fig. We then followed the same procedure as with cortical magnification: we summed the number of cycles across eccentricity bands, divided by the area of the visual field stimulated (1.5° to 9.5° eccentricity,

360° polar angle), and calculated the square root to yield a single summary measure per observer in units of cycles per deg –overall preferred spatial frequency.

## Supporting information

**S1 Text.  Three anisotropies in spatial frequency tuning.**
(DOCX)

**S1 Table.  Group-average preferred spatial frequency (cycles/deg) for individual V1 segments.** Each polar angle bin is defined as 22.5° of angle centered at the labeled value (i.e., 11.25° of angle includes data between 0° and 22.5° of polar angle). 0° of angle is defined as at the right horizontal meridian of the visual field and angle increases counterclockwise around the visual field. Each eccentricity bin is defined as 1° of eccentricity centered at the labeled value (i.e., 2° eccentricity includes data between 1.5°–2.5° eccentricity). These values are derived from log-Gaussians fit to beta weights from the 'combined' stimuli condition (i.e., the average of the beta weights for each pinwheel-annulus stimulus pair.
(XLSX)

**S2 Table.  Group-average V1 cortical magnification (mm/deg) for individual V1 segments.** Each polar angle bin is defined as 22.5° of angle centered at the labeled value (i.e., 11.25° of angle includes data between 0° and 22.5° of polar angle). 0° of angle is defined as at the right horizontal meridian of the visual field and angle increases counterclockwise around the visual field. Each eccentricity bin is defined as 1° of eccentricity centered at the labeled value (i.e., 2° eccentricity includes data between 1.5°–2.5° eccentricity).
(XLSX)

**S3 Table.  Median model parameter estimates and 95% confidence intervals from the models fit to individual participant measurements of preferred spatial frequency and cortical magnification.** 95% CIs are derived from bootstrapping across 40 observers.
(XLSX)

**S1 Fig.  Polar angle asymmetries in preferred spatial frequency vary with stimulus orientation.** (A) Combined condition; preferred spatial frequency is highest along the horizontal, intermediate along the lower vertical, and lowest along the upper vertical meridian. (B) Pinwheel stimuli; the polar angle asymmetries are boosted as the pinwheel stimuli contain horizontal content along the horizontal meridian and vertical content along the vertical meridian. (C) Annulus stimuli; the polar angle asymmetries are weakened. The data are fit with an inverse linear function from [7]. Error bars represent ±1 standard deviation (SD) across 50 bootstrapped group-averages.
(TIF)

**S2 Fig.  Polar angle asymmetries in V1 cortical magnification.** Cortical magnification plotted as a function of eccentricity for the horizontal meridian (HM: average of left and right horizontal), lower vertical (LVM), and upper vertical meridian (UVM). Data come from 22.5° wedge-ROIs centered either side of each meridian. The cortical magnification function from [7] is fit to the data from each meridian. Error bars represent ±1 SD across 50 bootstrapped group-averages.
(TIF)

**S3 Fig.  V1 spatial frequency bandwidth as a function of eccentricity.** Bandwidth (defined as σ of the log-Gaussian curve) varies as a function of eccentricity when measured along the horizontal, lower vertical, and upper vertical meridian, and all of V1 (combined stimulus condition). The meridian data are derived from 22.5° wedge-ROIs centered on either side of each meridian. All V1 data are averaged around polar angle. Bandwidth is indexed in octaves due to the logarithmic scaling of spatial frequency encoding in the visual system. Error bars represent ±1 standard deviation (SD) across 50 bootstrapped group-averages.
(TIF)

**S4 Fig. Comparing preferred spatial frequency as a function of eccentricity for scaled and uniform gratings.** Preferred spatial frequency is plotted as a function of eccentricity from two prior studies and current work. The current study and Broderick et. al. [11] fit the data with an inverse linear function, $f(r) = \dfrac{A}{r+B}$. We digitized the V1 data from Aghajari et. al. [12] Fig 4B and fit the same inverse linear function to their data. The three functions have slightly different shapes, however the estimates of preferred spatial frequency are close across the three studies.
(TIF)

**S5 Fig. Preferred spatial frequency as a function of eccentricity for individual observers.** Each colored line represents the change in V1 preferred spatial frequency as a function of eccentricity for an individual observer (n = 40).
(TIF)

## Acknowledgments

This work was supported in part through the NYU IT High Performance Computing resources, services, and staff expertise. We thank Ekin Tünçok and Shutian Xue for feedback on the manuscript.

## Author contributions

**Conceptualization:** Marisa Carrasco, Jonathan Winawer.

**Data curation:** Marc M. Himmelberg.

**Formal analysis:** Marc M. Himmelberg, Yuna Kwak, Jonathan Winawer.

**Funding acquisition:** Marc M. Himmelberg, Yuna Kwak, Marisa Carrasco, Jonathan Winawer.

**Investigation:** Marc M. Himmelberg.

**Methodology:** Marc M. Himmelberg, Jonathan Winawer.

**Project administration:** Marc M. Himmelberg.

**Supervision:** Jonathan Winawer.

**Visualization:** Marc M. Himmelberg.

**Writing – original draft:** Marc M. Himmelberg.

**Writing – review & editing:** Marc M. Himmelberg, Yuna Kwak, Marisa Carrasco, Jonathan Winawer.

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
