## [Decision Letter · Decision Letter 0]

14 Jul 2025

PCOMPBIOL-D-25-00961

Unpacking the V1 Map: Differential covariation of visual properties across spatial dimensions

PLOS Computational Biology

Dear Dr. Himmelberg,

Thank you for submitting your manuscript to PLOS Computational Biology. After careful consideration, we feel that it has merit but does not fully meet PLOS Computational Biology's publication criteria as it currently stands. Therefore, we invite you to submit a revised version of the manuscript that addresses the points raised during the review process.

Please submit your revised manuscript within 30 days Sep 13 2025 11:59PM. If you will need more time than this to complete your revisions, please reply to this message or contact the journal office at ploscompbiol@plos.org. Please include the following items when submitting your revised manuscript:

We look forward to receiving your revised manuscript.

Kind regards,

Paul Bays

Academic Editor

PLOS Computational Biology

Hugues Berry

Section Editor

PLOS Computational Biology

**Additional Editor Comments:**

All three reviewers were positive about your manuscript but have raised some suggestions for improvement that we ask you to address in a revision.

**Journal Requirements:**

At this stage, the following Authors/Authors require contributions: Marc Himmelberg, Yuna Kwak, Marisa Carrasco, and Jonathan Winawer. Please ensure that the full contributions of each author are acknowledged in the "Add/Edit/Remove Authors" section of our submission form.

5) We notice that your supplementary Figures, and Tables are included in the manuscript file. Please remove them and upload them with the file type 'Supporting Information'. Please ensure that each Supporting Information file has a legend listed in the manuscript after the references list.

6) Please ensure that the funders and grant numbers match between the Financial Disclosure field and the Funding Information tab in your submission form. Note that the funders must be provided in the same order in both places as well.

**Reviewers' comments:**

Reviewer's Responses to Questions

**Comments to the Authors:**

Reviewer #1: It was a pleasure reading this manuscript -- it is very well written, and I look forward to being able to use it as a teaching tool. Impact outside our field might not be great, but for anyone looking to quantify variation in perceptual properties across the visual field, this manuscript is both an excellent summary of the state of things and an exciting new dataset that informs our understanding of variation in perceptual phenomena (including some nice consideration of the origin of this variation).

I have no major concerns; just minor concerns listed in the order encountered in the manuscript.

I think the authors should address, head on and early on, the inherent circularity of comparing spatial frequency and cortical magnification, given the stimuli used. Both the radial and the polar gratings varied in spatial frequency with eccentricity, so the spatial frequency driving each voxel was calculated from that voxel's eccentricity. It is no surprise, then, if preferred spatial frequency depends on eccentricity. Similarly, a relationship to cortical magnification is guaranteed by computing spatial frequency after knowing a voxel's eccentricity.

Fig. 3 caption, line 138 and several other places: how is degrees^2 calculated? I'm assuming it is solid angle -- there really isn't another definition of square degrees, and the numbers look like solid angle numbers. But it's not obvious to someone who hasn't done this kind of work before, and the ms would benefit from 2 or 3 sentences in the Methods specifying that this is solid angle and providing the method for calculating it.

The last sentence in the Fig. 5 caption doesn't make sense until you get to the supplemental material and start trying to puzzle out the differences between Fig. 5 and Fig. S4. A few more words would help.

In Figs. 7 and 9, the labels to the left of the y-axis label are distracting and might not add to the intelligibility of the figure.

Line 314. Is "contrived" supposed to be a different word?

Line 541: "two" is duplicated

Supp. Fig S1 doesn't really have any context but is intriguing. It would be helpful to have some explanation of why bandwidth was computed and what relationship it has to the findings.

Cheryl Olman, PhD

University of Minnesota

Reviewer #2: This study uses fMRI in 40 participants to map how two key features of human primary visual cortex (V1), cortical magnification and preferred spatial frequency, covary across the visual field. The authors show that both decline proportionally with eccentricity, but this relationship breaks down across polar angle, where cortical magnification changes much more than spatial frequency. Larger V1 surface area predicts higher spatial frequency preferences. Analyses are robust, employing quantitative modeling and bootstrapping to confirm that these relationships differ between the two spatial axes, revealing dimension-specific V1 organization.

I have a few suggestions

In Figure 8, individual differences are assessed globally, i.e., overall V1 surface area vs. overall spatial frequency preference, collapsing across both eccentricity and polar angle. So regional effects may be missed. It would strengthen the study to analyze and/or discuss whether these relationships vary by polar angle.

While the link between cortical magnification and spatial frequency is quantified for eccentricity and linked to retinal ganglion cell density (Figure 9), the mechanisms for polar angle variation are less clear. Comparing V1 properties to retinal ganglion cell density across polar angle could clarify whether these asymmetries reflect retinal input or cortical processes.

This is entirely optional, but the current title is accurate yet somewhat generic. A more specific title that names cortical magnification, spatial frequency, and the axes of interest (eccentricity, polar angle) would better reflect the paper’s content.

Reviewer #3: This is a very interesting manuscript. I have only minor comments:

1. In figure 3 the error bars are noted to represent ±1 standard deviation

across 50 bootstrapped group-averages -- would it be possible to specify a bit more how the bootstrapping procedure was performed? how many subjects per group etc (this information is I think present later in the manuscript but it would be good to be present here also)

2. the variation along the polar angle is modeled as a factor that multiplies the average cortical magnification at each eccentricity. Would it be possible to comment whether there are deviations from this assumption? ie how accurate is this assumption?

3. The authors discuss the implications of their findings for behavioral experiments and suggest that there is good agreement with respect to the variation of spatial frequency/contrast sensitivity as a function of eccentricity. In contrast reports of differences observed along the polar angle are mixed, perhaps because the difference in the typical contrasts of stimuli used for fMRI imaging (high) versus for contrast sensitivity measurements (low). It would strengthen the paper considerably if the authors could implement a psychophysical task to test their observations. However, it is ok with me if this appears in a follow up manuscript.

**Have the authors made all data and (if applicable) computational code underlying the findings in their manuscript fully available?**

Reviewer #1: Yes

Reviewer #2: Yes

Reviewer #3: Yes

PLOS authors have the option to publish the peer review history of their article (what does this mean?). If published, this will include your full peer review and any attached files.

Reviewer #1: **Yes: ** Cheryl Olman

Reviewer #2: No

Reviewer #3: No

**Figure resubmission:**
---

## [Decision Letter · Decision Letter 1]

6 Oct 2025

Dear Dr Himmelberg,

We are pleased to inform you that your manuscript 'Unpacking the V1 map: Differential covariation of preferred spatial frequency and cortical magnification across spatial dimensions' has been provisionally accepted for publication in PLOS Computational Biology.

Best regards,

Paul Bays

Academic Editor

PLOS Computational Biology

Hugues Berry

Section Editor

PLOS Computational Biology

Reviewer's Responses to Questions

**Comments to the Authors:**

Reviewer #1: The authors have addressed all of my concerns from the previous review.

**Have the authors made all data and (if applicable) computational code underlying the findings in their manuscript fully available?**

Reviewer #1: Yes

PLOS authors have the option to publish the peer review history of their article (what does this mean?). If published, this will include your full peer review and any attached files.

Reviewer #1: **Yes: ** Cheryl A Olman

---

## [Editor Report · Acceptance letter]

PCOMPBIOL-D-25-00961R1

Unpacking the V1 map: Differential covariation of preferred spatial frequency and cortical magnification across spatial dimensions

Dear Dr Himmelberg,

I am pleased to inform you that your manuscript has been formally accepted for publication in PLOS Computational Biology. Your manuscript is now with our production department and you will be notified of the publication date in due course.

With kind regards,

Anita Estes
